# Bayesian Estimation of Latently-grouped Parameters in Undirected Graphical Models

**Jie Liu**
Dept of CS, University of Wisconsin
Madison, WI 53706
jieliu@cs.wisc.edu

**David Page**
Dept of BMI, University of Wisconsin
Madison, WI 53706
page@biostat.wisc.edu

## Abstract

In large-scale applications of undirected graphical models, such as social networks and biological networks, similar patterns occur frequently and give rise to similar parameters. In this situation, it is beneficial to group the parameters for more efficient learning. We show that even when the grouping is unknown, we can infer these parameter groups during learning via a Bayesian approach. We impose a Dirichlet process prior on the parameters. Posterior inference usually involves calculating intractable terms, and we propose two approximation algorithms, namely a Metropolis-Hastings algorithm with auxiliary variables and a Gibbs sampling algorithm with "stripped" Beta approximation (Gibbs_SBA). Simulations show that both algorithms outperform conventional maximum likelihood estimation (MLE). Gibbs_SBA's performance is close to Gibbs sampling with exact likelihood calculation. Models learned with Gibbs_SBA also generalize better than the models learned by MLE on real-world Senate voting data.

## 1 Introduction

Undirected graphical models, a.k.a. Markov random fields (MRFs), have many real-world applications such as social networks and biological networks. In these large-scale networks, similar kinds of relations can occur frequently and give rise to repeated occurrences of similar parameters, but the grouping pattern among the parameters is usually unknown. For a social network example, suppose that we collect voting data over the last 20 years from a group of 1,000 people who are related to each other through different types of relations (such as family, co-workers, classmates, friends and so on), but the relation types are usually unknown. If we use a binary pairwise MRF to model the data, each binary node denotes one person's vote, and two nodes are connected if the two people are linked in the social network. Eventually we want to estimate the pairwise potential functions on edges, which can provide insights about how the relations between people affect their decisions. This can be done via standard maximum likelihood estimation (MLE), but the latent grouping pattern among the parameters is totally ignored, and the model can be overparametrized. Therefore, two questions naturally arise. *Can MRF parameter learners automatically identify these latent parameter groups during learning? Will this further abstraction make the model generalize better, analogous to the lessons we have learned from hierarchical modeling [9] and topic modeling [5]?*

This paper shows that it is feasible and potentially beneficial to identify the latent parameter groups during MRF parameter learning. Specifically, we impose a Dirichlet process prior on the parameters to accommodate our uncertainty about the number of the parameter groups. Posterior inference can be done by Markov chain Monte Carlo with proper approximations. We propose two approximation algorithms, a Metropolis-Hastings algorithm with auxiliary variables and a Gibbs sampling algorithm with stripped Beta approximation (Gibbs_SBA). Algorithmic details are provided in Section 3 after we review related parameter estimation methods in Section 2. In Section 4, we evaluate our Bayesian estimates and the classical MLE on different models, and both algorithms outperform classical MLE. The Gibbs_SBA algorithm performs very close to the Gibbs sampling algorithm with exact likelihood calculation. Models learned with Gibbs_SBA also generalize better than the models learned by MLE on real-world Senate voting data in Section 5. We finally conclude in Section 6.

## 2 Maximum Likelihood Estimation and Bayesian Estimation for MRFs

Let $\mathcal{X} = \{0, 1, ..., m-1\}$ be a discrete space. Suppose that we have an MRF defined on a random vector $\mathbf{X} \in \mathcal{X}^d$ described by an undirected graph $\mathcal{G}(\mathcal{V}, \mathcal{E})$ with $d$ nodes in the node set $\mathcal{V}$ and $r$ edges in the edge set $\mathcal{E}$. The probability of one sample $\mathbf{x}$ from the MRF parameterized by $\boldsymbol{\theta}$ is

$$P(\mathbf{x}; \boldsymbol{\theta}) = \tilde{P}(\mathbf{x}; \boldsymbol{\theta})/Z(\boldsymbol{\theta}), \tag{1}$$

where $Z(\boldsymbol{\theta})$ is the partition function. $\tilde{P}(\mathbf{x}; \boldsymbol{\theta}) = \prod_{c \in \mathcal{C}(\mathcal{G})} \phi_c(\mathbf{x}; \boldsymbol{\theta}_c)$ is some unnormalized measure, and $\mathcal{C}(\mathcal{G})$ is some subset of cliques in $\mathcal{G}$, and $\phi_c$ is the potential function defined on the clique $c$ parameterized by $\boldsymbol{\theta}_c$. In this paper, we consider binary pairwise MRFs for simplicity, i.e. $\mathcal{C}(\mathcal{G}) = \mathcal{E}$ and $m=2$. We also assume that each potential function $\phi_c$ is parameterized by one parameter $\theta_c$, namely $\phi_c(\mathbf{X}; \theta_c) = \theta_c^{\mathbb{I}(X_u = X_v)}(1 - \theta_c)^{\mathbb{I}(X_u \neq X_v)}$ where $\mathbb{I}(X_u = X_v)$ indicates whether the two nodes $u$ and $v$ connected by edge $c$ take the same value, and $0 < \theta_c < 1, \forall c = 1, ..., r$. Thus, $\boldsymbol{\theta} = \{\theta_1, ..., \theta_r\}$. Suppose that we have $n$ independent samples $\mathbb{X} = \{\mathbf{x}^1, ..., \mathbf{x}^n\}$ from (1), and we want to estimate $\boldsymbol{\theta}$.

**Maximum Likelihood Estimate**: The MLE of $\boldsymbol{\theta}$ maximizes the log-likelihood function $\mathcal{L}(\boldsymbol{\theta}|\mathbb{X})$ which is concave w.r.t. $\boldsymbol{\theta}$. Therefore, we can use gradient ascent to find the global maximum of the likelihood function and find the MLE of $\boldsymbol{\theta}$. The partial derivative of $\mathcal{L}(\boldsymbol{\theta}|\mathbb{X})$ with respect to $\theta_i$ is $\frac{\partial \mathcal{L}(\boldsymbol{\theta}|\mathbb{X})}{\partial \theta_i} = \frac{1}{n} \sum_{j=1}^n \psi_i(\mathbf{x}^j) - E_{\boldsymbol{\theta}} \psi_i = E_{\mathbb{X}} \psi_i - E_{\boldsymbol{\theta}} \psi_i$ where $\psi_i$ is the sufficient statistic corresponding to $\theta_i$ after we rewrite the density into the exponential family form, and $E_{\boldsymbol{\theta}} \psi_i$ is the expectation of $\psi_i$ with respect to the distribution specified by $\boldsymbol{\theta}$. However the exact computation of $E_{\boldsymbol{\theta}} \psi_i$ takes time exponential in the treewidth of $\mathcal{G}$. A few sampling-based methods have been proposed, with different ways of generating particles and computing $E_{\boldsymbol{\theta}} \psi$ from the particles, including MCMC-MLE [11, 34], particle-filtered MCMC-MLE [1], contrastive divergence [15] and its variations such as persistent contrastive divergence (PCD) [29] and fast PCD [30]. Note that contrastive divergence is related to pseudo-likelihood [4], ratio matching [17, 16], and together with other MRF parameter estimators [13, 31, 12] can be unified as minimum KL contraction [18].

**Bayesian Estimate**: Let $\pi(\boldsymbol{\theta})$ be a prior of $\boldsymbol{\theta}$; then its posterior is $P(\boldsymbol{\theta}|\mathbb{X}) \propto \pi(\boldsymbol{\theta})\tilde{P}(\mathbb{X}; \boldsymbol{\theta})/Z(\boldsymbol{\theta})$. The Bayesian estimate of $\boldsymbol{\theta}$ is its posterior mean. Exact sampling from $P(\boldsymbol{\theta}|\mathbb{X})$ is known as doubly-intractable for general MRFs [21]. If we use the Metropolis-Hastings algorithm, then Metropolis-Hastings ratio is

$$a(\boldsymbol{\theta}^*|\boldsymbol{\theta}) = \frac{\pi(\boldsymbol{\theta}^*)\tilde{P}(\mathbb{X}; \boldsymbol{\theta}^*)Q(\boldsymbol{\theta}|\boldsymbol{\theta}^*)/Z(\boldsymbol{\theta}^*)}{\pi(\boldsymbol{\theta})\tilde{P}(\mathbb{X}; \boldsymbol{\theta})Q(\boldsymbol{\theta}^*|\boldsymbol{\theta})/Z(\boldsymbol{\theta})}, \tag{2}$$

where $Q(\boldsymbol{\theta}^*|\boldsymbol{\theta})$ is some proposal distribution from $\boldsymbol{\theta}$ to $\boldsymbol{\theta}^*$, and with probability $\min\{1, a(\boldsymbol{\theta}^*|\boldsymbol{\theta})\}$ we accept the move from $\boldsymbol{\theta}$ to $\boldsymbol{\theta}^*$. The real hurdle is that we have to evaluate the intractable $Z(\boldsymbol{\theta})/Z(\boldsymbol{\theta}^*)$ in the ratio. In [20], Møller *et al.* introduce one auxiliary variable $\mathbf{y}$ on the same space as $\mathbf{x}$, and the state variable is extended to $(\boldsymbol{\theta}, \mathbf{y})$. They set the new proposal distribution for the extended state $Q(\boldsymbol{\theta}, \mathbf{y}|\boldsymbol{\theta}^*, \mathbf{y}^*) = Q(\boldsymbol{\theta}|\boldsymbol{\theta}^*)\tilde{P}(\mathbf{y}; \boldsymbol{\theta})/Z(\boldsymbol{\theta})$ to cancel $Z(\boldsymbol{\theta})/Z(\boldsymbol{\theta}^*)$ in (2). Therefore by ignoring $\mathbf{y}$, we can generate the posterior samples of $\boldsymbol{\theta}$ via Metropolis-Hastings. Technically, this auxiliary variable approach requires perfect sampling [25], but [20] pointed out that other simpler Markov chain methods also work with the proviso that it converges adequately to the equilibrium distribution.

## 3 Bayesian Parameter Estimation for MRFs with Dirichlet Process Prior

In order to model the latent parameter groups, we impose a Dirichlet process prior on $\boldsymbol{\theta}$, which accommodates our uncertainty about the number of groups. Then, the generating model is

$$\begin{aligned} G &\sim \mathrm{DP}(\alpha_0, G_0) \\ \theta_i|G &\sim G, i = 1, ..., r \\ \mathbf{x}^j|\boldsymbol{\theta} &\sim F(\boldsymbol{\theta}), j = 1, ..., n, \end{aligned} \tag{3}$$

where $F(\boldsymbol{\theta})$ is the distribution specified by (1). $G_0$ is the base distribution (e.g. $\mathrm{Unif}(0, 1)$), and $\alpha_0$ is the concentration parameter. With probability 1.0, the distribution $G$ drawn from $\mathrm{DP}(\alpha_0, G_0)$ is discrete, and places its mass on a countably infinite collection of atoms drawn from $G_0$. In this model, $\mathbb{X} = \{\mathbf{x}^1, ..., \mathbf{x}^n\}$ is observed, and we want to perform posterior inference for $\boldsymbol{\theta} = (\theta_1, \theta_2, ..., \theta_r)$,

and regard its posterior mean as its Bayesian estimate. We propose two Markov chain Monte Carlo (MCMC) methods. One is a Metropolis-Hastings algorithm with auxiliary variables, as introduced in Section 3.1. The second is a Gibbs sampling algorithm with stripped Beta approximation, as introduced in Section 3.2. In both methods, the state of the Markov chain is specified by two vectors, $\mathbf{c}$ and $\boldsymbol{\phi}$. In vector $\mathbf{c} = (c_1, ..., c_r)$, $c_i$ denotes the group to which $\theta_i$ belongs. $\boldsymbol{\phi} = (\phi_1, ..., \phi_k)$ records the $k$ distinct values in $\{\theta_1, ..., \theta_r\}$ with $\phi_{c_i} = \theta_i$ for $i = 1, ..., r$. This way of specifying the Markov chain is more efficient than setting the state variable directly to be $(\theta_1, \theta_2, ..., \theta_r)$ [22].

## 3.1 Metropolis-Hastings (MH) with Auxiliary Variables

In the MH algorithm (see Algorithm 1), the initial state of the Markov chain is set by performing $K$-means clustering on MLE of $\boldsymbol{\theta}$ (e.g. from the PCD algorithm [29]) with $K = \lfloor \alpha_0 \ln r \rfloor$. The Markov chain resembles Algorithm 5 in [22], and it is ergodic. We move the Markov chain forward for $T$ steps. In each step, we update $\mathbf{c}$ first and then update $\boldsymbol{\phi}$. We update each element of $\mathbf{c}$ in turn; when resampling $c_i$, we fix $\mathbf{c}_{-i}$, all elements in $\mathbf{c}$ other than $c_i$. When updating $c_i$, we repeatedly for $M$ times propose a new value $c_i^*$ according to proposal $Q(c_i^*|c_i)$ and accept the move with probability $\min\{1, a(c_i^*|c_i)\}$ where $a(c_i^*|c_i)$ is the MH ratio. After we update every element of $\mathbf{c}$ in the current iteration, we draw a posterior sample of $\boldsymbol{\phi}$ according to the current grouping $\mathbf{c}$. We iterate $T$ times, and get $T$ posterior samples of $\boldsymbol{\theta}$. Unlike the tractable Algorithm 5 in [22], we need to introduce auxiliary variables to bypass MRF's intractable likelihood in two places, namely calculating the MH ratio (in Section 3.1.1) and drawing samples of $\boldsymbol{\phi}|\mathbf{c}$ (in Section 3.1.2).

### 3.1.1 Calculating Metropolis-Hastings Ratio

The MH ratio of proposing a new value $c_i^*$ for $c_i$ according to proposal $Q(c_i^*|c_i)$ is

$$
\begin{aligned}
a(c_i^*|c_i) &= \frac{\pi(c_i^*, \mathbf{c}_{-i}) P(\mathbb{X}; \boldsymbol{\theta}_{.i}^*) Q(c_i|c_i^*)}{\pi(c_i, \mathbf{c}_{-i}) P(\mathbb{X}; \boldsymbol{\theta}) Q(c_i^*|c_i)} \\
&= \frac{\pi(c_i^*|\mathbf{c}_{-i}) \tilde{P}(\mathbb{X}; \boldsymbol{\theta}_{.i}^*) Q(c_i|c_i^*)/Z(\boldsymbol{\theta}_{.i}^*)}{\pi(c_i|\mathbf{c}_{-i}) \tilde{P}(\mathbb{X}; \boldsymbol{\theta}) Q(c_i^*|c_i)/Z(\boldsymbol{\theta})},
\end{aligned}
$$

where $\boldsymbol{\theta}_{.i}^*$ is the same as $\boldsymbol{\theta}$ except its $i$-th element is replaced with $\phi_{c^*}$. The conditional prior $\pi(c_i^*|\mathbf{c}_{-i})$ is

$$
\pi(c_i = c|\mathbf{c}_{-i}) = \begin{cases} \frac{n_{-i,c}}{r-1+\alpha_0}, & \text{if } c \in \mathbf{c}_{-i} \\ \frac{\alpha_0}{r-1+\alpha_0}, & \text{if } c \notin \mathbf{c}_{-i} \end{cases}
$$

where $n_{-i,c}$ is the number of $c_j$ for $j \neq i$ and $c_j = c$. We choose proposal $Q(c_i^*|c_i)$ to be the conditional prior $\pi(c_i^*|\mathbf{c}_{-i})$, and the Metropolis-Hastings ratio can be further simplified as

---

**Algorithm 1** The Metropolis-Hastings algorithm

**Input:** observed data $\mathbb{X} = \{\mathbf{x}^1, ..., \mathbf{x}^n\}$

**Output:** $\hat{\boldsymbol{\theta}}^{(1)}, ..., \hat{\boldsymbol{\theta}}^{(T)}$; $T$ samples of $\boldsymbol{\theta}|\mathbb{X}$

**Procedure:**
Perform PCD algorithm to get $\tilde{\boldsymbol{\theta}}$, MLE of $\boldsymbol{\theta}$
Init. $\mathbf{c}$ and $\boldsymbol{\phi}$ via $K$-means on $\tilde{\boldsymbol{\theta}}$; $K = \lfloor \alpha_0 \ln r \rfloor$
**for** $t = 1$ **to** $T$ **do**
  **for** $i = 1$ **to** $r$ **do**
    **for** $l = 1$ **to** $M$ **do**
      Draw a candidate $c_i^*$ from $Q(c_i|c_i^*)$
      If $c_i^* \notin \mathbf{c}$, draw a value for $\phi_{c_i}$ from $G_0$
      Set $c_i = c_i^*$ with prob $\min\{1, a(c_i^*|c_i)\}$
    **end for**
  **end for**
  Draw a posterior sample of $\boldsymbol{\phi}$ according to current $\mathbf{c}$, and set $\hat{\theta}_i^{(t)} = \phi_{c_i}$ for $i = 1, ..., r$.
**end for**

---

$a(c_i^*|c_i) = \tilde{P}(\mathbb{X}; \boldsymbol{\theta}_{.i}^*) Z(\boldsymbol{\theta})/\tilde{P}(\mathbb{X}; \boldsymbol{\theta}) Z(\boldsymbol{\theta}_{.i}^*)$. However, $Z(\boldsymbol{\theta})/Z(\boldsymbol{\theta}_{.i}^*)$ is intractable. Similar to [20], we introduce an auxiliary variable $\mathbb{Z}$ on the same space as $\mathbb{X}$, and the state variable is extended to $(\mathbf{c}, \mathbb{Z})$. When proposing a move, we propose $c_i^*$ first and then propose $\mathbb{Z}^*$ with proposal $P(\mathbb{Z}; \boldsymbol{\theta}_{.i}^*)$ to cancel the intractable $Z(\boldsymbol{\theta})/Z(\boldsymbol{\theta}_{.i}^*)$. We set the target distribution of $\mathbb{Z}$ to be $P(\mathbb{Z}; \tilde{\boldsymbol{\theta}})$ where $\tilde{\boldsymbol{\theta}}$ is some estimate of $\boldsymbol{\theta}$ (e.g. from PCD [29]). Then, the MH ratio with the auxiliary variable is

$$
a(c_i^*, \mathbb{Z}^*|c_i, \mathbb{Z}) = \frac{P(\mathbb{Z}^*; \tilde{\boldsymbol{\theta}}) \tilde{P}(\mathbb{X}; \boldsymbol{\theta}_{.i}^*) \tilde{P}(\mathbb{Z}; \boldsymbol{\theta})}{P(\mathbb{Z}; \tilde{\boldsymbol{\theta}}) \tilde{P}(\mathbb{X}; \boldsymbol{\theta}) \tilde{P}(\mathbb{Z}^*; \boldsymbol{\theta}_{.i}^*)} = \frac{\tilde{P}(\mathbb{Z}^*; \tilde{\boldsymbol{\theta}}) \tilde{P}(\mathbb{X}; \boldsymbol{\theta}_{.i}^*) \tilde{P}(\mathbb{Z}; \boldsymbol{\theta})}{\tilde{P}(\mathbb{Z}; \tilde{\boldsymbol{\theta}}) \tilde{P}(\mathbb{X}; \boldsymbol{\theta}) \tilde{P}(\mathbb{Z}^*; \boldsymbol{\theta}_{.i}^*)}.
$$

Thus, the intractable computation of the MH ratio is replaced by generating particles $\mathbb{Z}^*$ and $\mathbb{Z}$ under $\boldsymbol{\theta}_{.i}^*$ and $\boldsymbol{\theta}$ respectively. Ideally, we should use perfect sampling [25], but it is intractable for general MRFs. As a compromise, we use standard Gibbs sampling with long runs to generate these particles.

### 3.1.2 Drawing Posterior Samples of $\boldsymbol{\phi}|\mathbf{c}$

We draw posterior samples of $\boldsymbol{\phi}$ under grouping $\mathbf{c}$ via the MH algorithm, again following [20]. The state of the Markov chain is $\boldsymbol{\phi}$. The initial state of the Markov chain is set by running PCD [29] with

parameters tied according to $\mathbf{c}$. The proposal $Q(\boldsymbol{\phi}^*|\boldsymbol{\phi})$ is a $k$-variate Gaussian $\mathcal{N}(\boldsymbol{\phi}, \sigma_Q^2 I_k)$ where $\sigma_Q^2 I_k$ is the covariance matrix. The auxiliary variable $\mathbb{Y}$ is on the same space as $\mathbb{X}$, and the state is extended to $(\boldsymbol{\phi}, \mathbb{Y})$. The proposal distribution for the extended state variable is $Q(\boldsymbol{\phi}, \mathbb{Y}|\boldsymbol{\phi}^*, \mathbb{Y}^*) = Q(\boldsymbol{\phi}|\boldsymbol{\phi}^*)\tilde{P}(\mathbb{Y}; \boldsymbol{\phi})/Z(\boldsymbol{\phi})$. We set the target distribution of $\mathbb{Y}$ to be $P(\mathbb{Y}; \tilde{\boldsymbol{\phi}})$ where $\tilde{\boldsymbol{\phi}}$ is some estimate of $\boldsymbol{\phi}$ such as the estimate from the PCD algorithm [29]. Then, the MH ratio for the extended state is

$$a(\boldsymbol{\phi}^*, \mathbb{Y}^*|\boldsymbol{\phi}, \mathbb{Y}) = \mathbb{I}(\boldsymbol{\phi}^* \in \boldsymbol{\Theta}) \frac{\tilde{P}(\mathbb{Y}^*; \tilde{\boldsymbol{\phi}})\tilde{P}(\mathbb{X}; \boldsymbol{\phi}^*)\tilde{P}(\mathbb{Y}; \boldsymbol{\phi})}{\tilde{P}(\mathbb{Y}; \tilde{\boldsymbol{\phi}})\tilde{P}(\mathbb{X}; \boldsymbol{\phi})\tilde{P}(\mathbb{Y}^*; \boldsymbol{\phi}^*)},$$

where $\mathbb{I}(\boldsymbol{\phi}^* \in \boldsymbol{\Theta})$ indicates that every dimension of $\boldsymbol{\phi}^*$ is in the domain of $G_0$. We set the state to be the new values with probability $\min\{1, a(\boldsymbol{\phi}^*, \mathbb{Y}^*|\boldsymbol{\phi}, \mathbb{Y})\}$. We move the Markov chain for $S$ steps, and get $S$ samples of $\boldsymbol{\phi}$ by ignoring $\mathbb{Y}$. Eventually we draw one sample from them randomly.

### 3.2 Gibbs Sampling with Stripped Beta Approximation

In the Gibbs sampling algorithm (see Algorithm 2), the initialization of the Markov chain is exactly the same as in the MH algorithm in Section 3.1. The Markov chain resembles Algorithm 2 in [22] and it can be shown to be ergodic. We move the Markov chain forward for $T$ steps. In each of the $T$ steps, we update $\mathbf{c}$ first and then update $\boldsymbol{\phi}$. When we update $\mathbf{c}$, we fix the values in $\boldsymbol{\phi}$, except we may add one new value to $\boldsymbol{\phi}$ or remove a value from $\boldsymbol{\phi}$. We update each element of $\mathbf{c}$ in turn. When we update $c_i$, we first examine whether $c_i$ is unique in $\mathbf{c}$. If so, we remove $\phi_{c_i}$ from $\boldsymbol{\phi}$ first. We then update $c_i$ by assigning it to an existing group or a new group with a probability proportional to a product of two quantities, namely

---

**Algorithm 2** The Gibbs sampling algorithm

**Input:** observed data $\mathbb{X} = \{\mathbf{x}^1, \mathbf{x}^2, ..., \mathbf{x}^n\}$
**Output:** $\hat{\boldsymbol{\theta}}^{(1)}, ..., \hat{\boldsymbol{\theta}}^{(T)}$; $T$ posterior samples of $\boldsymbol{\theta}|\mathbb{X}$
**Procedure:**
Perform PCD algorithm to get MLE $\tilde{\boldsymbol{\theta}}$
Init. $\mathbf{c}$ and $\boldsymbol{\phi}$ via $K$-means on $\tilde{\boldsymbol{\theta}}$; $K=\lfloor \alpha_0 \ln r \rfloor$
**for** $t = 1$ **to** $T$ **do**
  **for** $i = 1$ **to** $r$ **do**
    If current $c_i$ is unique in $\mathbf{c}$, remove $\phi_{c_i}$ from $\boldsymbol{\phi}$
    Update $c_i$ according to (4).
    If new $c_i \notin \mathbf{c}$, draw a value for $\phi_{c_i}$ and add to $\boldsymbol{\phi}$
  **end for**
  Draw a posterior sample of $\boldsymbol{\phi}$ according to current
  $\mathbf{c}$, and set $\hat{\theta}_i^{(t)} = \phi_{c_i}$ for $i = 1, ..., r$
**end for**

---

$$P(c_i = c|\mathbf{c}_{-i}, \mathbb{X}, \phi_{\mathbf{c}_{-i}}) \propto \begin{cases} \frac{n_{-i,c}}{r-1+\alpha_0} P(\mathbb{X}; \phi_c, \phi_{\mathbf{c}_{-i}}), \text{if } c \in \mathbf{c}_{-i} \\ \frac{\alpha_0}{r-1+\alpha_0} \int P(\mathbb{X}; \theta_i, \phi_{\mathbf{c}_{-i}}) \, dG_0(\theta_i), \text{if } c \notin \mathbf{c}_{-i}. \end{cases} \quad (4)$$

The first quantity is $n_{-i,c}$, the number of members already in group $c$. For starting a new group, the quantity is $\alpha_0$. The second quantity is the likelihood of $\mathbb{X}$ after assigning $c_i$ to the new value $c$ conditional on $\phi_{\mathbf{c}_{-i}}$. When considering a new group, we integrate the likelihood w.r.t. $G_0$. After $c_i$ is resampled, it is either set to be an existing group or a new group. If a new group is assigned, we draw a new value for $\phi_{c_i}$, and add it to $\boldsymbol{\phi}$. After updating every element of $\mathbf{c}$ in the current iteration, we draw a posterior sample of $\boldsymbol{\phi}$ under the current grouping $\mathbf{c}$. In total, we run $T$ iterations, and get $T$ posterior samples of $\boldsymbol{\theta}$. This Gibbs sampling algorithm involves two intractable calculations, namely (i) calculating $P(\mathbb{X}; \phi_c, \phi_{\mathbf{c}_{-i}})$ and $\int P(\mathbb{X}; \theta_i, \phi_{\mathbf{c}_{-i}}) \, dG_0(\theta_i)$ in (4) and (ii) drawing posterior samples for $\boldsymbol{\phi}$. We use a stripped Beta approximation in both places, as in Sections 3.2.1 and 3.2.2.

#### 3.2.1 Calculating $P(\mathbb{X}; \phi_c, \phi_{\mathbf{c}_{-i}})$ and $\int P(\mathbb{X}; \theta_i, \phi_{\mathbf{c}_{-i}}) \, dG_0(\theta_i)$ in (4)

In Formula (4), we evaluate $P(\mathbb{X}; \phi_c, \phi_{\mathbf{c}_{-i}})$ for different $\phi_c$ values with $\phi_{\mathbf{c}_{-i}}$ fixed and $\mathbb{X} = \{\mathbf{x}^1, \mathbf{x}^2, ..., \mathbf{x}^n\}$ observed. For ease of notation, we rewrite this quantity as a likelihood function of $\theta_i$, $\mathcal{L}(\theta_i|\mathbb{X}, \boldsymbol{\theta}_{-i})$, where $\boldsymbol{\theta}_{-i} = \{\theta_1, ..., \theta_{i-1}, \theta_{i+1}, ..., \theta_r\}$ is fixed. Suppose that the edge $i$ connects variables $X_u$ and $X_v$, and we denote $\mathbf{X}_{-uv}$ to be the variables other than $X_u$ and $X_v$. Then

$$\mathcal{L}(\theta_i|\mathbb{X}, \boldsymbol{\theta}_{-i}) = \prod_{j=1}^n P(x_u^j, x_v^j|\mathbf{x}_{-uv}^j; \theta_i, \boldsymbol{\theta}_{-i}) P(\mathbf{x}_{-uv}^j; \theta_i, \boldsymbol{\theta}_{-i})$$

$$\approx \prod_{j=1}^n P(x_u^j, x_v^j|\mathbf{x}_{-uv}^j; \theta_i, \boldsymbol{\theta}_{-i}) P(\mathbf{x}_{-uv}^j; \boldsymbol{\theta}_{-i}) \propto \prod_{j=1}^n P(x_u^j, x_v^j|\mathbf{x}_{-uv}^j; \theta_i, \boldsymbol{\theta}_{-i}).$$

Above we approximate $P(\mathbf{x}_{-uv}^j; \theta_i, \boldsymbol{\theta}_{-i})$ with $P(\mathbf{x}_{-uv}^j; \boldsymbol{\theta}_{-i})$ because the density of $\mathbf{X}_{-uv}$ mostly depends on $\boldsymbol{\theta}_{-i}$. The term $P(\mathbf{x}_{-uv}^j; \boldsymbol{\theta}_{-i})$ can be dropped since $\boldsymbol{\theta}_{-i}$ is fixed, and we only have

to consider $P(x_u^j, x_v^j | \mathbf{x}_{-uv}^j; \theta_i, \boldsymbol{\theta}_{-i})$. Since $\boldsymbol{\theta}_{-i}$ is fixed and we are conditioning on $\mathbf{x}_{-uv}^j$, they together can be regarded as a fixed potential function telling how likely the rest of the graph thinks $X_u$ and $X_v$ should take the same value. Suppose that this fixed potential function (the message from the rest of the network $\mathbf{x}_{-uv}^j$) is parameterized as $\eta_i$ ($0 < \eta_i < 1$). Then

$$\prod_{j=1}^{n} P(x_u^j, x_v^j | \mathbf{x}_{-uv}^j; \theta_i, \boldsymbol{\theta}_{-i}) \propto \prod_{j=1}^{n} \lambda^{\mathbb{I}(x_u^j = x_v^j)} (1-\lambda)^{\mathbb{I}(x_u^j \neq x_v^j)} = \lambda^{\sum_{j=1}^{n} \mathbb{I}(x_u^j = x_v^j)} (1-\lambda)^{\sum_{j=1}^{n} \mathbb{I}(x_u^j \neq x_v^j)} \quad (5)$$

where $\lambda = \theta_i \eta_i / \{\theta_i \eta_i + (1-\theta_i)(1-\eta_i)\}$. The end of (5) resembles a Beta distribution with parameters $(\sum_{j=1}^{n} \mathbb{I}(x_u^j = x_v^j) + 1, n - \sum_{j=1}^{n} \mathbb{I}(x_u^j = x_v^j) + 1)$ except that only part of $\lambda$, namely $\theta_i$, is random. Now we want to use a Beta distribution to approximate the likelihood with respect to $\theta_i$, and we need to remove the contribution of $\eta_i$ and only consider the contribution from $\theta_i$. We choose $\text{Beta}(\lfloor n\tilde{\theta}_i \rfloor + 1, n - \lfloor n\tilde{\theta}_i \rfloor + 1)$ where $\tilde{\theta}_i$ is MLE of $\theta_i$ (e.g. from the PCD algorithm). This approximation is named the *Stripped Beta Approximation*. The simulation results in Section 4.2 indicate that the performance of the stripped Beta approximation is very close to using exact calculation. Also this approximation only requires as much computation as in the tractable tree-structure MRFs, and it does not require generating expensive particles as in the MH algorithm with auxiliary variables. The integral $\int P(\mathbb{X}; \theta_i, \phi_{\mathbf{c}_{-i}}) \, dG_0(\theta_i)$ in (4) can be calculated via Monte Carlo approximation. We draw a number of samples of $\theta_i$ from $G_0$, and evaluate $P(\mathbb{X}; \theta_i, \phi_{\mathbf{c}_{-i}})$ and take the average.

### 3.2.2 Drawing Posterior Samples of $\phi | \mathbf{c}$

The stripped Beta approximation also allows us to draw posterior samples from $\phi | \mathbf{c}$ approximately. Suppose that there are $k$ groups according to $\mathbf{c}$, and we have estimates for $\phi$, denoted as $\hat{\phi} = (\hat{\phi}_1, ..., \hat{\phi}_k)$. We denote the numbers of elements in the $k$ groups by $\mathbf{m} = \{m_1, ..., m_k\}$. For group $i$, we draw a posterior sample for $\phi_i$ from $\text{Beta}(\lfloor m_i n \hat{\phi}_i \rfloor + 1, m_i n - \lfloor m_i n \hat{\phi}_i \rfloor + 1)$.

## 4 Simulations

We investigate the performance of our Bayesian estimators on three models: (i) a tree-MRF, (ii) a small grid-MRF whose likelihood is tractable, and (iii) a large grid-MRF whose likelihood is intractable. We first set the ground truth of the parameters, and then generate training and testing samples. On training data, we apply our grouping-aware Bayesian estimators and two baseline estimators, namely a grouping-blind estimator and an oracle estimator. The grouping-blind estimator does not know groups exist in the parameters, and estimates the parameters in the normal MLE fashion. The oracle estimator knows the ground truth of the groupings, and ties the parameters from the same group and estimates them via MLE. For the tree-MRF, our Bayesian estimator is exact since the likelihood is tractable. For the small grid-MRF, we have three variations for the Bayesian estimator, namely Gibbs sampling with exact likelihood computation, MH with auxiliary variables, and Gibbs sampling with stripped Beta approximation. For the large grid-MRF, the computational burden only allows us to apply Gibbs sampling with stripped Beta approximation.

We compare the estimators by three measures. The first is the average absolute error of estimate $1/r \sum_{i=1}^{r} |\theta_i - \hat{\theta}_i|$ where $\hat{\theta}_i$ is the estimate of $\theta_i$. The second measure is the log likelihood of the testing data, or the log pseudo-likelihood [4] of the testing data when exact likelihood is intractable. Thirdly, we evaluate how informative the grouping yielded by the Bayesian estimator is. We use the *variation of information* metric [19] between the inferred grouping $\hat{C}$ and the ground truth grouping $C$, namely $\text{VI}(\hat{C}, C)$. Since $\text{VI}(\hat{C}, C)$ is sensitive to the number of groups in $\hat{C}$, we contrast it with $\text{VI}(\bar{C}, C)$ where $\bar{C}$ is a random grouping with its number of groups the same as $\hat{C}$. Eventually, we evaluate $\hat{C}$ via the VI difference, namely $\text{VI}(\bar{C}, C) - \text{VI}(\hat{C}, C)$. A larger value of VI difference indicates a more informative grouping yielded by our Bayesian estimator. Because we have one grouping in each of the $T$ MCMC steps, we average the VI difference yielded in each of the $T$ steps.

### 4.1 Simulations on Tree-structure MRFs

For the structure of the MRF, we choose a perfect binary tree of height 12 (i.e. 8,191 nodes and 8,190 edges). We assume there are 25 groups among the 8,190 parameters. The base distribution $G_0$ is $\text{Unif}(0, 1)$. We first generate the true parameters for the 25 groups from $\text{Unif}(0, 1)$. We then randomly assign each of the 8,190 parameters to one of the 25 groups. We then generate 1,000

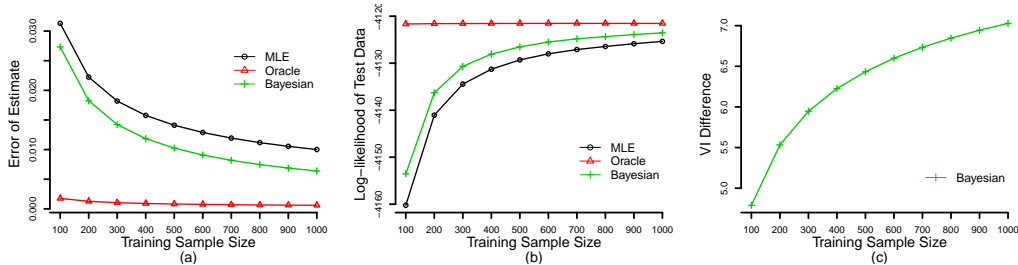

Figure 1: Performance of the grouping-blind MLE, the oracle MLE and our Bayesian estimator on tree-structure MRFs in terms of (a) error of estimate and (b) log-likelihood of test data. Subfigure (c) shows the VI difference between the grouping yielded by our Bayesian estimator and random grouping.

testing samples and $n$ training samples ($n=100, 200, ..., 1,000$). Eventually, we apply the grouping-blind MLE, the oracle MLE, and our grouping-aware Bayesian estimator on the training samples. For tree-structure MRFs, both MLE and Bayesian estimation have a closed form solution. For the Bayesian estimator, we set the number of Gibbs sampling steps to be 500 and set $\alpha_0=1.0$. We replicate the experiment 500 times, and the averaged results are in Figure 1.

Our grouping-aware Bayesian estimator has a lower estimate error and a higher log likelihood of test data, compared with the grouping-blind MLE, demonstrating the "blessing of abstraction". Our Bayesian estimator performs worse than oracle MLE, as we expect. In addition, as the training sample size increases, the performance of our Bayesian estimator approaches that of the oracle

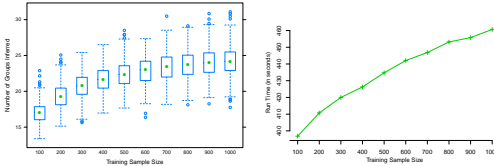

Figure 2: Number of groups inferred by the Bayesian estimator and its run time.

MLE. The VI difference in Figure 1(c) indicates that the Bayesian estimator also recovers the latent grouping to some extent, and the inferred groupings become more and more reliable as the training size increases. The number of groups inferred by the Bayesian estimator and its running time are in Figure 2. We also investigate the asymptotic performance of the estimators and their performance when there are no parameter groups. The results are provided in the supplementary materials.

## 4.2 Simulations on Small Grid-MRFs

For the structure of the MRF, we choose a $4\times4$ grid with 16 nodes and 24 edges. Exact likelihood is tractable in this small model, which allows us to investigate how good the two types of approximation are. We apply the grouping-blind MLE (the PCD algorithm), the oracle MLE (the PCD algorithm with the parameters from same group tied) and three Bayesian estimators: Gibbs sampling with exact likelihood computation (Gibbs_ExactL), Metropolis-Hastings with auxiliary variables (MH_AuxVar), and Gibbs sampling with stripped Beta approximation (Gibbs_SBA). We assume there are five parameter groups. The base distribution is $\text{Unif}(0, 1)$. We first generate the true parameters for the five groups from $\text{Unif}(0, 1)$. We then randomly assign each of the 24 parameters to one of the five groups. We then generate 1,000 testing samples and $n$ training samples ($n=100, 200, ..., 1,000$). For Gibbs_ExactL and Gibbs_SBA, we set the number of Gibbs sampling steps to be 100. For MH_AuxVar, we set the number of MH steps to be 500 and its proposal number $M$ to be 5. The parameter $\sigma_Q$ in Section 3.1.2 is set to be 0.001 and the parameter $S$ is set to be 100. For all three Bayesian estimators, we set $\alpha_0=1.0$. We replicate the experiment 50 times, and the averaged results are in Figure 4.

Our grouping-aware Bayesian estimators have a lower estimate error and a higher log likelihood of test data, compared with the grouping-blind MLE, demonstrating the blessing of abstraction. All three Bayesian estimators perform worse than oracle MLE, as we expect. The VI difference in Figure 4(c) indicates that the

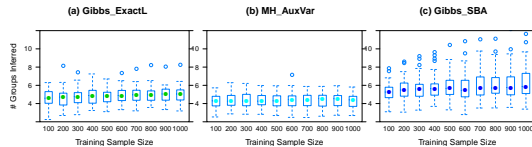

Figure 3: The number of groups inferred by Gibbs_ExactL, MH_AuxVar and Gibbs_SBA.

Bayesian estimators also recover the grouping to some extent, and the inferred groupings become more and more reliable as the training size increases. In Figure 3, we provide the boxplots of the number of groups inferred by Gibbs_ExactL, MH_AuxVar and Gibbs_SBA. All three methods recover a reasonable number of groups, and Gibbs_SBA slightly over-estimates the number of groups.

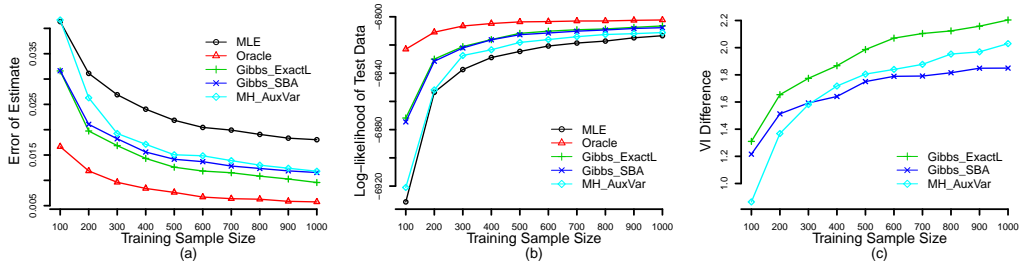

Figure 4: Performance of grouping-blind MLE, oracle MLE, Gibbs_ExactL, MH_AuxVar, and Gibbs_SBA on the small grid-structure MRFs in terms of (a) error of estimate and (b) log-likelihood of test data. Subfigure (c) shows the VI difference between the grouping yielded by our Bayesian estimators and random grouping.

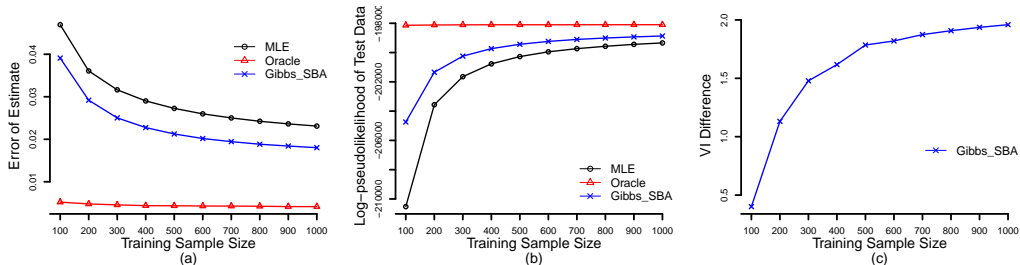

Figure 5: Performance of the grouping-blind MLE, the oracle MLE and the Bayesian estimator (Gibbs_SBA) on large grid-structure MRFs in terms of (a) error of estimate and (b) log-likelihood of test data. Subfigure (c) shows the VI difference between the grouping yielded by our Bayesian estimator and random grouping.

Among the three Bayesian estimators, Gibbs_ExactL has the lowest estimate error and the highest log likelihood of test data. Gibbs_SBA also performs considerably well, and its performance is close to the performance of Gibbs_ExactL. MH_AuxVar works slightly worse, especially when there is less training data.

Table 1: The run time (in seconds) of Gibbs_ExactL, MH_AuxVar and Gibbs_SBA when training size is $n$.

|  | $n$=100 | $n$=500 | $n$=1,000 |
|---|---|---|---|
| GIBBS_EXACTL | 88,136.3 | 91,055.0 | 92,503.4 |
| MH_AUXVAR | 540.2 | 3,342.2 | 4,546.7 |
| GIBBS_SBA | 8.1 | 10.8 | 14.2 |

However, MH_AuxVar recovers better groupings than Gibbs_SBA when there are more training data. The run times of the three Bayesian estimators are listed in Table 1. Gibbs_ExactL has a computational complexity that is exponential in the dimensionality $d$, and cannot be applied to situations when $d > 20$. MH_AuxVar is also computationally intensive because it has to generate expensive particles. Gibbs_SBA runs fast, with its burden mainly from running PCD under a specific grouping in each Gibbs sampling step, and it scales well.

### 4.3 Simulations on Large Grid-MRFs

The large grid consists of 30 rows and 30 columns (i.e. 900 nodes and 1,740 edges). Exact likelihood is intractable for this large model, and we cannot run Gibbs_ExactL. The high dimension also prohibits MH_AuxVar. Therefore, we only run the Gibbs_SBA algorithm on this large grid-structure MRF. We assume that there are 10 groups among the 1,740 parameters. We also evaluate the estimators by the log pseudo-likelihood of testing data. The other settings of the experiments stay the same as Section 4.2. We replicate the experiment 50 times, and the averaged results are in Figure 5.

For all 10 training sets, our Bayesian estimator Gibbs_SBA has a lower estimate error and a higher log likelihood of test data, compared with the grouping-blind MLE (via the PCD algorithm). Gibbs_SBA has a higher estimate error and a lower pseudo-likelihood of test data than the oracle MLE. The VI difference in Figure 5(c) indicates that Gibbs_SBA gradually recovers the grouping as the training size increases.

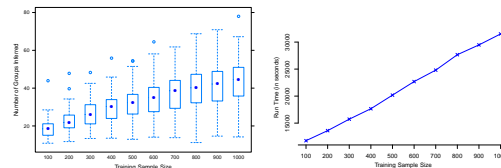

Figure 6: Number of groups inferred by Gibbs_SBA and its run time.

The number of groups inferred by Gibbs_SBA and its running time are provided in Figure 6. Similarly to the observation in Section 4.2, Gibbs_SBA overestimates the number of groups. Gibbs_SBA finishes the simulations on 900 nodes and 1,740 edges in hundreds of minutes (depending on the training size), which is considered to be very fast.

Table 2: Log pseudo-likelihood (LPL) of training and testing data from MLE (PCD) and Bayesian estimate (Gibbs_SBA), the number of groups inferred by Gibbs_SBA, and its run time in the Senate voting experiments.

| | LPL-TRAIN | | LPL-TEST | | | |
| | MLE | GIBBS_SBA | MLE | GIBBS_SBA | # GROUPS | RUNTIME (MINS) |
|---|---|---|---|---|---|---|
| EXP1 | -10716.75 | -10721.34 | -9022.01 | -8989.87 | 7.89 | 204 |
| EXP2 | -8306.17 | -8322.34 | -11490.47 | -11446.45 | 7.29 | 183 |

## 5 Real-world Application

We apply the Gibbs_SBA algorithm on US Senate voting data from the 109th Congress (available at www.senate.gov). The 109th Congress has two sessions, the first session in 2005 and the second session in 2006. There are 366 votes and 278 votes in the two sessions, respectively. There are 100 senators in both sessions, but Senator Corzine only served the first session and Senator Menendez only served the second session. We remove them. In total, we have 99 senators in our experiments, and we treat the votes from the 99 senators as the 99 variables in the MRF. We only consider contested votes, namely we remove the votes with less than ten or more than ninety supporters. In total, there are 292 votes and 221 votes left in the two sessions, respectively. The structure of the MRF is from Figure 13 in [2]. There are in total 279 edges. The votes are coded as $-1$ for no and $1$ for yes. We replace all missing votes with $-1$, staying consistent with [2]. We perform two experiments. First, we train the MRF using the first session data, and test on the second session data. Then, we train on the second session and test on the first session. We compare our Bayesian estimator (via Gibbs_SBA) and MLE (via PCD) by the log pseudo-likelihood of testing data since exact likelihood is intractable. We set the number of Gibbs sampling steps to be 3,000. Both of the two experiments are finished in around three hours on a single CPU. The results are summarized in Table 2. In the first experiment, the log pseudo-likelihood of test data is $-9022.01$ from MLE, whereas it is $-8989.87$ from our Bayesian estimate. In the second experiment, the log pseudo-likelihood of test data is $-11490.47$ from MLE, whereas it is $-11446.45$ from our Bayesian estimate. The increase of log pseudo-likelihood is comparable to the increase of log (pseudo-)likelihood we gain in the simulations (please refer to Figures 1b, 4b and 5b at the points when we simulate 200 and 300 training samples). Both experiments indicate that the models trained with the Gibbs_SBA algorithm generalize considerably better than the models trained with MLE. Gibbs_SBA also infers there are around eight different types of relations among the senators. The two trained models are provided in the supplementary materials, and the estimated parameters in the two models are consistent.

## 6 Discussion

Bayesian nonparametric approaches [23, 10], such as the Dirichlet process [7], provide an elegant way of modeling mixtures with an unknown number of components. These approaches have yielded advances in different machine learning areas, such as the infinite Gaussian mixture models [26], the infinite mixture of Gaussian processes [27], infinite HMMs [3, 8], infinite HMRFs [6], DP-nonlinear models [28], DP-mixture GLMs [14], infinite SVMs [33, 32], and the infinite latent attribute models [24]. In this paper, we play the same trick of replacing the prior distribution with a prior stochastic process to accommodate our uncertainty about the number of parameter groups. To the best of our knowledge, this is the first time a Bayesian nonparametric approach is applied to models whose likelihood is intractable. Accordingly, we propose two types of approximation, namely a Metropolis-Hastings algorithm with auxiliary variables and a Gibbs sampling algorithm with stripped Beta approximation. Both algorithms show superior performance over conventional MLE, and Gibbs_SBA can also scale well to large-scale MRFs. The Markov chains in both algorithms are ergodic, but may not be in detailed balance because we rely on approximation. Thus, we guarantee that both algorithms converge for general MRFs, but they may not exactly converge to the target distribution.

In this paper, we only consider the situation where the potential functions are pairwise and there is only one parameter in each potential function. For graphical models with more than one parameter in the potential functions, it is appropriate to group the parameters on the level of potential functions. A more sophisticated base distribution $G_0$ (such as some multivariate distribution) needs to be considered. In this paper, we also assume the structures of the MRFs are given. When the structures are unknown, we still need to perform structure learning. Allowing structure learners to automatically identify structure modules will be another very interesting topic to explore in the future research.

**Acknowledgements**
The authors acknowledge the support of NIGMS R01GM097618-01 and NLM R01LM011028-01.

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
