[Supplementary Material]

# Supplementary Material for "Bayesian Estimation of Latently-grouped Parameters in Undirected Graphical Models"

Jie Liu, David Page

November 7, 2013

# 1   More Simulation Results

## 1.1   "Trace" of Bayesian Estimator, Suggested by Reviewer 2

The second reviewer recommended us to report the "trace" of the Bayesian estimator in terms of the number of Gibbs steps performed. We performed our Bayesian estimator on Simulation 1 and reported its performance after $l$ iterations, $l = 1$, 10, 25, 50, 100, 250, 500. We also test three sample sizes, namely $n = 100$, 500, 1,000. The performance is reported in Figure 1. It is observed that the Bayesian estimator converges after 100 iterations.

## 1.2   Performance of Clustered MLE, Suggested by Reviewer 2

The second reviewer recommended us to introduce another baseline, "clustered MLE". We performed K-Means algorithm on the MLE of the parameters (K is set to be the ground truth of group numbers) and performed MLE with the clusters. We termed this baseline MLE-KMeans. Its performance in the large-grid MRFs simulation is reported in Figure 2. It is observed that the performance of clustered MLE is comparable with standard grouping-blind MLE.

## 1.3   Large Sample Performance

In Section 4.1 of main text, we try 10 training set with different sizes, namely 100, 200, ..., and 1,000. We also empirically investigate the asymptotic property of the three estimators, namely the situation when we have a large number of training samples, such as 2,000, 5,000, 10,000 and 20,000 training samples. The simulation results are provided in Figure 3. It is observed that our Bayesian estimator still has an edge over the grouping blind MLE when there are sufficient data.

## 1.4   When There Are No Parameter Groups

In Section 4.1 of main text, we assume there exist 25 groups among the parameters in the tree-structure MRF model. What happens to our grouping-aware Bayesian estimator if there are no groups among the parameters? Here, we directly generate the 8,190 parameters from $U(0,1)$ rather than the DP prior, and then generate the training data and the testing data. The other settings of the experiments are the same as in the previous experiments. We again try the 10 different training sample sizes and replicate the experiment 500 times. There is no separate oracle estimator in this set of experiments, because the grouping-blind MLE serves as the oracle now. We compare the performance of the grouping-blind estimator and our grouping-aware estimator. The results are reported in Figure 4. Quite surprisingly, in all the 10 training sets of different sample sizes, the performance of our grouping-aware estimator is comparable with that of the grouping-blind estimator. The results tell us that even if we are provided with wrong prior knowledge about the existence

Figure 1: The "trace" of our Bayesian estimator on tree-structure MRFs in terms of (a) error of estimate and (b) log-likelihood of test data. Subfigure (c) shows the VI difference between the grouping yielded by our Bayesian estimator and random grouping.

Figure 2: Performance of the grouping-blind MLE, the clustered MLE (MLE-KMeans), the oracle MLE and our Bayesian estimator on tree-structure MRFs in terms of (a) error of estimate and (b) log-likelihood of test data. Subfigure (c) shows the VI difference between the grouping yielded by our Bayesian estimator and random grouping.

of parameter groups, our grouping-aware estimator barely harms the parameter estimation. It is also interesting to look at the number of groups yielded in the two situations (see Figure 5). When there are 25 groups among the parameters, the number of groups inferred by our algorithm is also around 25. When there is no group, the number of groups inferred by our algorithm is 19 given 100 training samples. The number of groups inferred by our estimator also gradually increases to 43 as the training sample size increases to 1,000. Also the entropy from the grouping yielded by our algorithm increases as the training sample size increases, as shown in subfigure 4(c). In other words, with more training data, our grouping-aware Bayesian estimator can gradually learn the truth about the parameters even when the prior knowledge is misleading.

## 2 The models learned on Senate voting records data

In Section 5 of main text, we performed two experiments. In the first experiment, we use the first session as the training data and use the second session as the testing data. In the second experiment, we use the second session as the training data and use the first session as the testing data. We report the two trained models along with estimated parameters in Figure 6 and 7. Note that the estimated parameters are not from several distinct values, and the reason is that our approach is Bayesian and the recovered groups are "soft" in the sense the parameters are only the same in one MCMC step.

Figure 3: Performance of the grouping-blind MLE, the oracle MLE and our Bayesian estimator on tree-structure MRFs in terms of (a) error of estimate and (b) log-likelihood of test data. Subfigure (c) shows the VI difference between the grouping yielded by our Bayesian estimator and random grouping.

Figure 4: Performance of the grouping-blind estimator (Oracle) and our grouping-aware estimator (Bayesian) on tree-structure MRFs in terms of (a) average estimate error and (b) log-likelihood of test data when there exist no groups among the parameters. Subfigure (c) shows the entropy of the grouping yielded by our algorithm.

Figure 5: Boxplots of the number of groups inferred by our algorithm in each of 500 replications when we vary the number of training samples, in the experiments (a) there indeed exist 25 groups among the parameters and (b) there are no groups among the parameters.

Figure 6: The trained MRF using the votes in the first session of 109th Congress.

Figure 7: The trained MRF using the votes in the second session of 109th Congress.