[Reviews · NeurIPS 2013]

Submitted by Assigned_Reviewer_5

The paper uses the intuition that the real data is often generated in a setting where different components share characteristics to suggest that if the data is to be modeled by a graphical model (MRF in the paper) then it is appropriate to assume that the parameter values should somehow come in clusters. To impose such clustering, they propose a DP-mixture prior over the parameters of the MRFs and work out several approximations to Bayesian learning of parameters for an MRF with a given structure.

The paper is well-structured and written, and the empirical evaluation on the simulated data is thorough. The proposed idea (clustering of the parameters of the MRFs) to the best of my knowledge is novel, and the paper is technically sound, at least based on my cursory pass through the details. There are enough details in the paper to reproduce the results (albeit the paper is understandably dense).

Having said all that, I do not necessarily see the improvements over the unconstrained MLE on real data (Senate voting) is due to the assumptions that the authors are basing their idea on. MRFs can be reformulated, perhaps with over-parameterization (e.g., a tree-structured MRF can be represented by a tree BN), and I do not see why the parameters under such reparameterization would exhibit discernible clusters. It is more likely that an unconstrained MLE would overfit, so imposing a strong prior (infinite mixture over parameters) yields a better fit. It would be interesting to see if a simple 2-component mixture prior would do even better than a more computationally complex DP mixture.

Minor:

Section 4 -- need to define VI.

Figure 2 is unreadable.
Summary: The paper proposes an interesting prior over the parameters of a graphical model and describes approximations to the Bayesian approach to learning the parameters.

Submitted by Assigned_Reviewer_6

The paper tackles the difficult problem of hierarchical parameter
estimation in Markov random fields. The model consists of a Dirichlet
process prior for parameters, and the computational contribution is an
approximate MCMC method, since ``exact'' MCMC seems somewhat out of
reach for this problem.

The solution is not particularly elegant, but to be fair the problem
is among the worst for MCMC samplers to tackle.

It would be interesting to understand which of the main features of
the solution contribute most for the results. To begin with, a trace
of the approximate MCMC progression is absent. Without the trace, this
makes me think that most of the work is done by the initialization
procedure and that the Bayesian aspect of it is
minimal. (Nevertheless, I still think that just making this work at
all is already not easy.) What is missing perhaps is a simple
comparison against the method hinted by the initialization procedure:
first, fit parameters; then cluster them; then refit parameters using
the clustered MLE objective function.

The write-up of the method in 3.2 could be improved. Notation gets in
the way. For example, in the equations of line 211-213 (p. 4), please
do not write it as P(X;, theta_i, theta_minus_i) as indicating it
proportional to the last element on that line makes no sense.
Instead, write something like L(theta_i; X, theta_minus_i) or some
other notation so that is is clearer what is random and what is not
here.

The intuitive justification for the beta approximation is a bit
disappointing, since one again it will rely on using PCD (and how is
theta_tilde_i obtained? By fixing c and the other thetas? Is c_i
marginalized? This needs to be clarified.)

I like the experiments comparing "full Bayesian" to MLE and others,
but I have to say I'm quite amazed how close the full Bayes and the
SBA are to each other. That alone make the paper interesting for
discussion, but as I said the thing missing is some sensitivity
analysis trying to understand better which of the aspects of the
approach are the reason for the success.

Minor: is there a way of scaling the vertical axis on the VI
Difference plots so we can understand how well the method does in
absolute?
Summary: A practical solution to a general and hard problem on learning multivariate distributions. Good in general, needs some sensitivity analysis on whether the true power behind it is simpler than what the approach proposes.

Submitted by Assigned_Reviewer_7

This paper proposes a Bayesian approach to learning binary pairwise MRFs with latently tied parameters. A DP prior is used to model the grouping of parameters, and two approximate inference methods are compared with MLE.

Doing Bayesian inference on MRFs is difficult, known as doubly intractable. The first algorithm, M-H with auxiliary variables, is based on the paper of Moller et.al. 2006. It is technically sound (except for the sample generated by Gibbs sampling), but could be slow, hard to mix, and therefore not feasible for real medium-sized applications. The second algorithm, Gibbs_SBA, is novel to my knowledge and seems to be a good candidate for approximate inference. The experiments show that the approximate methods are much faster than the exact method and obtain similar performance on small problems. It would be a meaningful contribution to Bayesian approach to MRFs.

My concern is on the usage of the Stripped Beta Approximation. The derivation of the method is lack of clarity. It is based on two approximations: (1) line 215: the distribution of all variables other than u and v is independent on the value of \theta_i; (2) line 228: approximate the contribution of \theta_i in (5) with a Beta distribution. I don’t understand how the second approximation is adopted. Apparently Equation 5 is a Beta function of \lambda, which is a complicated function of \theta. The authors don’t provide sufficient explanation for the beta approximation or the usage of the MLE of \theta in the Beta parameters. While the experiments show the overall performance of Gibbs_SBA is similar to the exact method in terms of the two first metrics, a figure or a paragraph of discussion on the accuracy of the approximation on one single step of sampling would be very useful.

In the experiment of the real problem, the output of the parameters does not show any hard grouping because the Bayesian approach assumes a posterior distribution. So is the latent grouping prior useful only as regularization to prevent over-fitting, or you can make more meanings out of it? The improvement on the pseudo-likelihood is not very significant. I guess that is because the sampling procedure is initialized on the result of PCD.

My last concern is about the speed. The computation burden is known as a serious problem for Bayesian inference. The authors compared the speed of the two approximate methods with the exact method. But how are they compared with the MLE algorithm based on PCD? It seems that for the fastest method for Bayesian inference, Gibbs_SBA, one has to run MLE (PCD) for every parameter on every iteration. Would it be applicable for any large problem in the real world?

Some minor questions and typos:
1. The binary pairwise MRF in the paper has pairwise potentials only. Would Gibbs_SBA still applies for an MRF with unary bias?
2. For M-H algorithm in section 3.1.1, do you also need to integrate over \theta when a new group is created?
3. Line 126: Section 3.2.2 -> 3.1.2
Summary: A Bayesian MRF model is proposed with latently-grouped parameters, and two approximate inference algorithms are discussed. The second algorithm, Gibbs_SBA, is a novel algorithm and could be good contribution for Bayesian inference on MRF, although the deviation or the validity is not clear.
Author Feedback

Author rebuttal: We appreciate the excellent advice from three reviewers. They provided many useful suggestions which can dramatically improve the paper. Below are our replies to the individual reviewers. The additional experiments and results (suggested by the reviewers) are provided in the supplementary material (we do not include them in the main text because of the space restriction).

Reviewer 1 (Assigned_Reviewer_5):

We thank you for your insightful review. We agree with you that on the real data, the improvement is from the strong prior on the parameters which provides further abstraction, just like what we gain during hierarchical modeling and topic modeling. Your idea of using "2-component mixture prior" on the real data is a smart way to diagnose. We just quickly implemented this idea (initialize with two clusters, and keeping the number of clusters to be 2 all through). Its LPL-TEST is -9094.11 in Exp1, and is -11616.89 in Exp2. Its LPL-TRAIN is -11206.38 in Exp1, and is -8618.72 in Exp2. It seemed to us that "2-component mixture prior" performs even worse than MLE. Therefore, we feel the DP part is useful to recover the number of hidden groups, and this is important.

We will define VI in the paper.

We will make Figure 2 more readable.


Reviewer 2 (Assigned_Reviewer_6):

We thank you for your insightful review. We like your suggestion on presenting the "trace" of the Bayesian estimator. We will report this result in the supplementary material (for a given training set, report the estimation error and likelihood of data in terms of the number of MCMC steps performed).

Your suggestion of adding "clustered MLE" is excellent. We implemented this baseline in our experiments, and the performance of clustered MLE is comparable with standard MLE. Therefore, we believe the true power behind our estimator is inferring the hidden groups under the Bayesian framework. We will report the corresponding results and add the discussion in the supplementary materials. We appreciate that you pointed this out.

Thanks for your suggestions (such as replace density function with likelihood function) on rewriting section 3.2. We will try to give more intuitive justification for the beta approximation. For your questions: you are right, theta_tilde_i is from PCD. When we evaluate the group of one parameter, namely c_i , all the rest of the groupings and parameters stay fixed. Then we set the new group of the parameter according to the likelihood, according to (4).

We do not think there is a way of scaling the vertical axis on the VI Difference plots in the literature.


Reviewer 3 (Assigned_Reviewer_7):

All your comments are exactly correct and insightful. There are exactly two approximations within the usage of the Stripped Beta Approximation. For the second approximation, the motivation comes from the fact that the Beta distribution is a conjugate prior to the binomial distribution. Suppose that on edge i, we have a potential function parameterized by theta_i. At the same time, let's imagine that the rest of network imposes another potential function parameterized by \eta_i, telling how likely the rest of the graph thinks X_u and X_v should take the same value (\eta_i stays fixed). When we multiply the two potentials, we get the one potential function parameterized by \lambda_i. However, when we evaluate the posterior distribution of \theta_i (conditional on the rest), we only observe the data (the count of X_u=X_v) which correspond to \lambda_i. Therefore, we have to remove the contribution from the rest of the network, namely "\eta_i". On the other hand, PCD provides an MLE estimator of \theta_i, which implicitly teases apart \theta_i and \eta_i. Therefore in the end, we can get an approximate posterior distribution for \theta_i based on the MLE of \theta_i from PCD. We like your suggestion of further providing "a figure or a paragraph of discussion on the accuracy of the approximation on one single step of sampling". We will do that.

I agree with your comments on the real problem. The improvement is from the strong prior on the parameters which provides further abstraction, just like what we gain during hierarchical modeling and topic modeling. There might be no direct way of evaluating whether the improvement on the pseudo-likelihood is significant or not, but we do feel the increase of log pseudo-likelihood is comparable to the increase of log (pseudo-)likelihood we gain in the simulations (please refer to Figures 1b, 4b and 5b at the points when we simulate 200 and 300 training samples).

You are exactly right that the computational burden is high because likelihood is intractable and Bayesian inference is usually computation intensive. The computational complexity is linear with the number of edges. We feel it should work fine for graphs with thousands of nodes (what you referred to as "medium-sized"). Please also note that our algorithm can be easily parallelized because it is basically MCMC.

For your minor comments:

1. Yes, our algorithm can handle unary bias (node potential) since we can learn the node potential with PCD.
2. We do not need to integrate over \theta when a new group is created in the MH algorithm. We choose proposal Q(c_i^*|c_i) to be the conditional prior \pi(c_i^*|{\bf c}_{-i}). If a new group is proposed, we draw a value for the new group from G_0. Our MH algorithm resembles the Algorithm 5 in [22] R. M. Neal. Markov chain sampling methods for Dirichlet process mixture models. Journal of Computational and Graphical Statistics, 9(2):249–265, 2000.
3. Thanks for pointing out the typo.